# Brown Algae as Functional Food Source of Fucoxanthin: A Review

**DOI:** 10.3390/foods11152235

**Published:** 2022-07-27

**Authors:** Nur Akmal Solehah Din, ‘Ain Sajda Mohd Alayudin, Noor-Soffalina Sofian-Seng, Hafeedza Abdul Rahman, Noorul Syuhada Mohd Razali, Seng Joe Lim, Wan Aida Wan Mustapha

**Affiliations:** 1Department of Food Sciences, Faculty of Science and Technology, Universiti Kebangsaan Malaysia, Bangi 43600, Selangor, Malaysia; nurakmalsolehah.din@gmail.com (N.A.S.D.); ainsajda@gmail.com (‘A.S.M.A.); soffalina@ukm.edu.my (N.-S.S.-S.); hafeedzarahman@ukm.edu.my (H.A.R.); syuhada_ns@ukm.edu.my (N.S.M.R.); joe@ukm.edu.my (S.J.L.); 2Innovation Centre for Confectionery Technology (MANIS), Faculty of Science and Technology, Universiti Kebangsaan Malaysia, Bangi 43600, Selangor, Malaysia

**Keywords:** fucoxanthin, carotenoid, aquatic natural product, brown seaweed, functional foods

## Abstract

Fucoxanthin is an algae-specific xanthophyll of aquatic carotenoid. It is prevalent in brown seaweed because it functions as a light-harvesting complex for algal photosynthesis and photoprotection. Its exceptional chemical structure exhibits numerous biological activities that benefit human health. Due to these valuable properties, fucoxanthin’s potential as a potent source for functional food, feed, and medicine is being explored extensively today. This article has thoroughly reviewed the availability and biosynthesis of fucoxanthin in the brown seaweed, as well as the mechanism behind it. We included the literature findings concerning the beneficial bioactivities of fucoxanthin such as antioxidant, anti-inflammatory, anti-obesity, antidiabetic, anticancer, and other potential activities. Last, an additional view on its potential as a functional food ingredient has been discussed to facilitate a broader application of fucoxanthin as a promising bioactive compound.

## 1. Introduction

Individuals around the world have long ranked health as their top priority. Food sources and dietary supplements are frequently important in achieving optimal health. Hence, natural bioactive compounds have become more sought after as functional food ingredients over the years, due to the wide range of health advantages they provide. Algae are one of the greatest sources of natural bioactive substances, and they are found abundantly in marine, freshwater, and even terrestrial environments. They can tolerate a broad range of pH, temperature, turbidity, as well as different concentrations of oxygen and carbon dioxide [1]. The size may vary from the microscopic unicellular microalgae to the macroscopic multicellular macroalgae, commonly known as seaweed, reaching up to more than 60 m in length [2].

Seaweed is the photosynthetic aquatic plant or algae commonly found in coastal regions that make up the base of most aquatic food chains—basically, the primary producer. They are high in nutritional values such as fatty acids, proteins, vitamins, minerals, and fibres, besides palatable as it has a pleasant umami taste [3]. The chemical composition of seaweed changes depending on species, maturity, habitat light intensity, salinity, temperature, and environments [4,5,6]. These aquatic macroalgae are categorised into three classes based on their predominant pigmentation: brown (Phaeophyceae), green (Chlorophyceae), and red (Rhodophyceae) seaweed [7]. Brown algae contain polysaccharides that are complex and have a high alcohol content. Its plant walls are made up of alginic acid, a biopolymer formed of hydrophilic polyuronic acid chains, and form viscous gum when hydrated [8]. About 2200 species of brown seaweed were found, and most of them grow in cold water. In contrast, red seaweed walls constituted long-chain polysaccharides, i.e., cellulose agar and carrageenan. About 6500 species of red seaweeds were known, mostly in the coral areas. Finally, the green seaweed is rich in chlorophyll compounds, with more than 7000 species reported in freshwater, marine habitats, and terrestrially grown in soil, trees, and rocks [9,10]. According to Badar, et al. [11], algae are often able to survive in harsh conditions with high levels of heavy metals and can be exploited as a source of energy, chemicals, and food.

In 2019, the global production of seaweed was approximately 35.8 million tonnes. The majority is from the Asian region which contributed 97% of the total production, including China, Korea, Japan, the Philippines, Malaysia, and Indonesia. World seaweed production is concentrated on 5 genera that account for about 95% of the world production: brown (35.4% *Laminaria*/*Saccharina*, 7.4% *Undaria*), red (33.5% *Kappaphycus/Eucheuma*, 10.5% *Gracilaria*, and 8.6% *Porphyra*), and none from green seaweed. Although seaweed production is still dependent on wild stock harvesting (68% of total production), seaweed cultivation has been developing recently, especially in several countries in Europe (32% of total production) [12]. The world seaweed cultivation has increased tremendously from 1990 to 2019, i.e., brown seaweed (3.1 to 16.4 million tonnes) and red seaweed (1.0 to 18.3 million tonnes). Nonetheless, green seaweed cultivation declined by approximately halved throughout the years (31 to 17 thousand tonnes) [13]. Generally, seaweeds have been prevalently known for their therapeutic and nutritional properties through their biomedical applications. Subsequently, its application is further developed in food sectors as phycocolloids, thickening, and stabilising gelling agents. Over the past decades, multiple studies have been performed to investigate the relevance and beneficial qualities of aquatic macroalgae seaweed. However, vast attention has been given to aquatic carotenoids in the past decade as they were reported to possess several potent biological activities.

Carotenoids as bioactive compounds are present in all algae, as they represent photosynthetic pigments responsible for the red, yellow, and orange colours. Some photosynthetic organisms utilise it as a critical part of the photosystem assembly and light-harvesting processes necessary for photosynthesis. Moreover, it has prominent photoprotective properties capable of inactive reactive oxygen species (ROS) that are generated when exposed to light and air [14]. Interestingly, carotenoids are precursors to several plant hormones such as abscisic acid, which is implicated in abiotic stress responses in plants (e.g., low temperature and/or dehydration), and the strigolactone phytohormone, which functions in inhibiting the shoot branching in the plant [15,16,17]. Carotenoids are a type of tetraterpene, a hydrocarbon with a linear C40 molecular backbone, and can contain up to 11 conjugated double bonds [18,19]. This compound can be further subclassified into two groups: (1) carotenes—pure hydrocarbons containing no oxygen—and (2) xanthophylls—oxygenated carotenes, which contain one or more oxygen molecules, where fucoxanthin is classified under xanthophylls [20]. Figure 1 summarises the carotenoids present in general seaweed.

Originating from the same group (carotenoids), xanthophylls and carotenes share common physical and chemical characteristics such as antioxidant and lipophilicity because of their capacity to reduce reactive nitrogen and oxygen species [21]. However, the existence of oxygen in xanthophylls, with the presence of many hydroxyl groups (OH) and keto groups (=O), causes it to be more polar and less hydrophobic than pure carotenes [22,23]. Xanthophylls components are categorised into two types: (1) primary xanthophyll, i.e., an essential cellular photosynthetic system of algae cells that is important for seaweed survival as they act as a light-harvesting antenna, and (2) secondary xanthophyll, i.e., metabolites that are made in large quantities after being exposed to certain environmental stimuli, such as carotenogenesis [24]. Fucoxanthin, a xanthophyll component, is known as a secondary metabolite produced in brown seaweed chloroplast [25]. The compound has been widely investigated due to its tremendous biological protective properties for human health and has been industrially produced from seaweeds. However, the demand for fucoxanthin surpasses its production, and biotechnological approaches are extensively multiplied to provide a sufficient supply of this high-quality compound via economical operations without adversely affecting nature.

## 2. Function and Availability of Fucoxanthin in Seaweed

In the main, fucoxanthin is found in brown seaweed, responsible for its brown to a yellow hue. The presence of fucoxanthin is very dominant in brown seaweed, overshadowing other pigments (i.e., chlorophyll-a, chlorophyll-c, β-carotene, and other xanthophylls). Meanwhile, the red seaweed has dominant red pigments from phycoerythrin and phycocyanin, while green seaweed contains chlorophyll-a and chlorophyll-b that give it the green colour [9]. Fucoxanthin contributes to more than 10% of the approximated natural production of carotenoids each year [21,26]. Generally, this compound functions as an effective plant photoprotection light-harvesting component, as well as initiating photosynthesis upregulation [27]. Fucoxanthin resides in the membrane-bound compartments (thylakoids) of the chloroplast, where it binds to chlorophyll a, c, and apoproteins, eventually creating light-harvesting complexes in the spectrum’s blue-green region and transferring energy to the photosynthetic electron transport chain in algae. Pyszniak and Gibbs [28] revealed that fucoxanthin alone could increase photosynthesis efficiency where a broader light spectrum (449–540 nm) was absorbed. Meanwhile, chlorophyll-a absorbs light strongly at light spectrums of 400–450 and 650–700 nm. Thus, fucoxanthin and chlorophyll were reported to be strongly complementary for the plant light absorption [29]. The production of macroalgae seaweed, especially brown seaweed, is extensively large due to the plenty of sources and cost-effective to be used commercially. Fucoxanthin is available from farmed or wild brown seaweed and can be extracted using several methods such as solvent extraction, Soxhlet-assisted extraction (SAE), enzyme-assisted extraction, ultrasound-assisted extraction, etc. [30]. From the methodological aspect, factors such as drying and extraction methods have been stated to significantly affect the fucoxanthin amount obtained from seaweed [30,31].

Generally, the content of fucoxanthin may vary across seaweed species and genus variations [32,33]. Moreover, environmental aspects such as geographical location and harvesting season have an impact on fucoxanthin content as they can affect photosynthetic activity [6]. Terasaki, et al. [34] reported that during winter, where there is limited access to sunlight, fucoxanthin concentration is upregulated for photosynthetic activity in brown seaweed. Correspondingly, it is higher during winter than summer as it is significantly affected by the physicochemical properties of seawater (pH, salinity, and conductivity) and the surrounding factors where the seaweed grows. For example, during summer, the high seawater temperature and prolonged sunlight exposure (>16 h) reduce the generation of fucoxanthin in seaweed. Karkhaneh Yousefi, et al. [35] stated that several external factors such as water nutritional profile and depth could interfere with fucoxanthin accumulation in seaweed. Additionally, the condition of the seaweed sample was reported to alter the concentration of fucoxanthin in seaweed. According to Zarekarizi, et al. [36], fucoxanthin content in dried seaweed may be less than the fresh one; however, several studies do not show such results. Table 1 shows the amount of fucoxanthin reported across various species of brown seaweed from several countries.

## 3. Biosynthesis of Fucoxanthin in Seaweed

Remarkably, fucoxanthin (Figure 2) possesses a peculiar structure due to the allenic bond, a 5,6-monoepoxide, nine conjugated double bonds, and acetyl groups attached to the polyene chain of the compound [69]. Around 40 of 700 naturally occurring carotenoids have the allenic bond. The allenic carotenoids are principally fucoxanthin from brown seaweed, metabolites of fucoxanthin (fucoxanthinol and amarouciaxanthin A), neoxanthin from higher plants, and perinjin from microalgae [70]. This structure distinguishes fucoxanthin from other seaweed carotenoids like β-carotene and lutein [18].

The carotenoid biosynthesis pathway in seaweed can further explain the formation of fucoxanthin and the allenic bond. Each seaweed taxonomical group (i.e., brown, green, and red seaweed) has different carotenoid formations at each biochemical stage, illustrated in Figure 3. Generally, only the carotenoid biosynthesis pathway of brown seaweed includes xanthophyll of fucoxanthin. In contrast, although the red and green seaweeds produce zeaxanthin and lutein, other xanthophylls such as violaxanthin and neoxanthin are not produced by the red seaweed. This shows that each seaweed contains class-specific compositions of xanthophyll.

Fucoxanthin synthesis depends on the products of two pathways, i.e., the mevalonate (MVA) pathway in the cytoplasm or the non-mevalonate l-deoxy-D-xylulose-5- phosphate/2-C-methyl-D-erythritol 4-phosphate (DOXP/MEP) pathway in plastids as precursors. The chemical breakdown of pyruvate by glycolysis and lipid by β-oxidation generated acetyl-CoA compounds in plants. Through the MVA pathway, acetyl-CoA is transformed to isopentenyl diphosphate (IPP), the main precursor for isoprenoid biosynthesis of carotenoids. In the absence of the MVA pathway, algae have developed efficient IPP synthesis through the MEP pathway [71].

In the lycopene biosynthesis pathway, the IPP molecule condensation by the geranylgeranyl pyrophosphate synthase (GGPPS) will result in the geranylgeranyl pyrophosphate (GGPP) formation, important for protein prenylation. This stage regulates several developmental processes in plants [17,72]. The condensation process of two GGPP (C_20_) molecules is the primary metabolism step, that is stimulated via phytoene synthase (PSY); considered a rate-limiting enzyme for entry reaction into the biosynthetic pathway [73,74,75]. In the next stage, the phytoene (C_40_) is generated and its conversion to lycopene generally includes four sets of desaturation processes [76,77]. The lycopene production in photosynthetic cells and the first two desaturation processes are catalysed by phytoene desaturase (PDS). Meanwhile, the other two processes involve ζ-carotene isomerase (ZISO) as well as ζ-carotene desaturase (ZDS) enzymes that cause the formation of neurosporene. Carotenoid isomerase (CRTISO) in parallel with ζ-carotene desaturation isomerises the cis-double bonds of neurosporene into all-trans-lycopene [17,78]. Each compound is sequentially modified throughout the lycopene biosynthesis, resulting in the increased number of conjugated carbon-carbon double bonds.

In the carotenoid biosynthesis pathway, the terminal isoprene structures of all-trans-lycopene molecules are then cyclised through the β-ionone ring formation to create β-carotene via lycopene β-cyclase (LCYB). Meanwhile, α-carotene is generated via the lycopene ε-cyclase (LCYE) catalytic action. In this step, the xanthophyll distribution varies across the different types of seaweed (green, red, and brown). Next, the carotene hydroxylation is mediated by β-carotene hydroxylase (BCH), producing zeinoxanthin and zeaxanthin in green and brown seaweeds. Both types of xanthophyll are present in the red seaweed pathway. However, its xanthophyll cycle is quite brief, besides the lack of compound biosynthesis after zeaxanthin, resulting in the zeaxanthin and lutein accumulation as major carotenoids [79,80,81]. Consequently, the green and brown seaweeds possess the violaxanthin cycle, which involves epoxidation or de-epoxidation, a reversible consecutive transformation of zeaxanthin, antheraxanthin, and violaxanthin. The reaction is part of the photosynthetic machinery to regulate the plant’s sunlight absorption under numerous ecological stress situations [18,82].

According to Mikami and Hosokawa [18], since brown seaweed originates from the red algae secondary endosymbiosis, and the LCYE enzyme gene appears to have been produced by gene duplication in algae, it is hypothesised to lose the gene of LCYE and develop novel genes for xanthophyll syntheses, e.g., fucoxanthin, diadinoxanthin, and diatoxanthin. Consequently, the brown seaweed pathway is observed to have another xanthophyll cycle, either the neoxanthin or diadinoxanthin cycles. The neoxanthin cycle involves the conversion of β-carotene to diadinoxanthin and fucoxanthin from neoxanthin. Neoxanthin is the compound where the pathway diverges, and its molecules have a similar configuration to fucoxanthin [83].

In the violaxanthin cycle, all modifications are symmetrical for carotenoid molecule terminals. However, asymmetrical modifications of the molecule can generate asymmetrical products like neoxanthin through allenic double bond formation at one terminal of the carotenoid. The entire process involves a proton-catalysed cleaving of the 5,6-epoxy aromatic ring and proton removal from C7, which results in the formation of a new double bond at C6 [84]. On the contrary, the allenic double bonds and acetylenic bonds have a strong biosynthetic connection during the conversion of neoxanthin to diadinoxanthin. The diadinoxanthin cycle generally is a reversible interconversion between diadinoxanthin and diatoxanthin, leading to the formation of fucoxanthin. According to Dambek, Eilers, Breitenbach, Steiger, Büchel and Sandmann [84], the formation of acetylenic bonds is derived from a proton-catalysed reaction of the existing allenic double bonds as well as the 5′-HO group removal as a water molecule. Concurrently, the fucoxanthin structure also includes the keto group formation at the other terminal of the molecule; C8 hydroxylation together with the C7 double bond tautomerisation. This compound is generated as a final product from both cycles with different biosynthesis pathways dependent on the enzyme availability in the algae.

## 4. Metabolism, Bioavailability and Bioaccessibility of Fucoxanthin

In the gastrointestinal tract, dietary fucoxanthin is promptly hydrolysed to fucoxanthinol by digestive enzymes of lipase and cholesterol esterase within two hours. Then, fucoxanthin absorption occurred in intestinal cells before being transformed to amarouciaxanthin A in the liver, as illustrated in Figure 4. In vitro study in Caco-2 cells exhibited the hydrolysis of fucoxanthin to fucoxanthinol and transformation to amarouciaxanthin A via dehydrogenation/isomerisation process in the microsomes of mice liver as well as HepG2 cells of human liver tissue [85,86]. It was found that dietary fucoxanthin favourably resides as a fucoxanthinol compound in most tissues and plasma whereas, amarouciaxanthin A compound located in abdominal white adipose tissue (WAT) of mice [87,88]. Thus, the bioavailability of its metabolites is much higher than fucoxanthin. Hashimoto, Ozaki, Taminato, Das, Mizuno, Yoshimura, Maoka and Kanazawa [87] have conducted a daily oral assessment of fucoxanthin, and it was found that only a minute quantity of fucoxanthin was not digested after a week of treatment. Most of the fucoxanthin metabolites (>80%, 11.486 µg/mg protein) were found in the adipose WAT, and approximately 2.127 µg/mg protein accumulated in other tissue lipids [89]. The high amount of fucoxanthin metabolite accumulation in the adipose tissue indicates it as the main target, possibly related to its anti-obesity and antidiabetic effects. Theoretically, fucoxanthin is tremendously vulnerable to transformation and degradation mainly in the gastrointestinal tract situation as acidic conditions of the stomach and gut enzymes such as lipase and cholesterol esterase can cause subsequent absorption of fucoxanthin. Generally, the bioavailability of fucoxanthin can be determined by its digestion, absorption, metabolism, distribution, and bioactivity.

Given the ratio of fucoxanthin adsorption to its dose in metabolite analysis, fucoxanthin metabolites of fucoxanthinol and amarouciaxanthin A have better levels of absorption than astaxanthin in mice [87]. Meanwhile, fucoxanthin was observed to be absorbed more efficiently than lutein esters in the mouse intestine, although both compounds have a similar mechanism of absorption [87,90,91,92]. Fucoxanthin absorbed in the intestinal mucosal epithelial cell of the small intestine will enter the lymph circulation system before being reduced to amorouciaxanthin A in the liver. According to Matsumoto, et al. [90], although fucoxanthin was not found in the lymph duct and vein, its metabolite of fucoxanthinol was identified in the lymph fluid, indicating its conversion to fucoxanthinol via lymphatic absorption occurred in the intestine. It is also probable that several constituents such as dietary fibre present in the seaweed matrix inhibit fucoxanthin intestinal absorption [91]. However, a study demonstrated the absorption of fucoxanthin without conversion to any metabolites [92].

In vitro studies using a simulated gastrointestinal digestion model are commonly used to evaluate the bioaccessibility of a bioactive compound, as they are simple, cheap, and repeatable [93,94,95,96]. Guo, et al. [97] studied the fucoxanthin’s bioaccessibility and deacetylation via in vitro simulation of the gastrointestinal digestion system and colonic batch fermentation. It was found that both fucoxanthin and fucoxanthinol were bioaccessibly higher than 32.7% after gastrointestinal digestion. But, a huge loss of fucoxanthin and production changes of short-chain fatty acid was detected during colon fermentation. The properties of fucoxanthin, which are non-polar, hydrophobic, and insoluble in water, can lead to low bioaccessibility in the body. Yet, fucoxanthin can form aggregates by intermolecular bonding, and microencapsulation has been reported as an effective technique to avoid the fast degradation of fucoxanthin and improve its bioaccessibility [98,99]. Fucoxanthin bioaccessibility is dependent on diet variations, i.e., affected by the composition of food and gut microbiota. Food matrices such as proteins or dietary fibres may bind to fucoxanthin and reduce its bioaccessibility in gastrointestinal fluids. It is decomposed, utilised by intestinal microorganisms, and eventually excreted via faeces [100]. A study by Liu, et al. [101] reported that the main microorganisms in the gut, such as *Escherichia coli* and *Lactobacillii*, can deacetylate fucoxanthin into fucoxanthinol and increase its bioaccessibility. Lastly, bioaccessibility of consumed fucoxanthin can be mediated via facilitated diffusion of scavenger receptor B1 (SR-B1) in the intestinal epithelium. This can cause a difference in fucoxanthin’s effective concentration in the circulatory system and eventually reduce its functional and beneficial result [102].

## 5. Potential Biological Activities of Fucoxanthin

The exceptional chemical structure of fucoxanthin exhibits a wide range of functionalities. Generally, fucoxanthin can occur in a cis or trans configuration. However, according to Holdt and Kraan [103], the trans isomer has better stability, comprising 90% of naturally found fucoxanthin and has more potent antioxidants than the cis isomer. The structure of the fucoxanthin allenic bond is unique and imparts functional health impacts such as antioxidant, anti-obesity, anti-inflammatory, and many more [47,100,104,105,106,107,108]. The ability of fucoxanthin to modulate and alter the biological system of individual gene and protein expression is reported to be the key mechanism for its physiological activities [104].

### 5.1. Antioxidant Activity

Carotenoids provide a protective mechanism against free radicals and singlet oxygen from causing harm to cells and tissues, which can be achieved by physical interaction through the transfer of excess energy of the singlet oxygen to the long central chain of fucoxanthin conjugated polyene structure. This causes molecule excitation to the triplet state where energy is released as heat, relaxing the molecule back to a singlet state—free from any structural change. Unlike other common carotenoids, fucoxanthin possesses antioxidant activity due to its allenic bond, hydroxyl groups of aromatic rings, and epoxide structure. This electron-rich structure contributes to the ability of fucoxanthin to scavenge various types of free radicals, for example, trichloromethylperoxy radical (CCl_3_ O_2_•), lipid-free radicals, hydroxyl radical (OH•), nitrogen dioxide radical (NO_2_•), sulfoxyl radicals (RSO_2_•), and many more. The fucoxanthin chain structure has been reported to have nine conjugated double bonds and several functional groups that can support the quenching ability of the singlet oxygen [105]. Numbers of studies presented the potential antioxidant activities of fucoxanthin based on several assays such as 2,2-diphenyl-1-picrylhydrazyl (DPPH), ferric reducing antioxidant power (FRAP), 2,2′-azinobis-3-ethyl benzothiazoline-6-sulfonate (ABTS), and others [38,109,110,111,112]. For example, fucoxanthin was detected to have an IC_50_ of 0.20 mg/mL in the DPPH experiment and a FRAP value of 65 mmol Fe^2 +^ /g. Furthermore, it has been shown to have dose-dependent antioxidant effects, with a 3.3-fold rise in the ratio of reduced to oxidised glutathione [106]. Recently, Zhao, et al. [107] has reported that fucoxanthin purified by silica gel column chromatography (SGCC) has exhibited strong antioxidant properties, with effective concentrations of 0.14 mgmL^−1^ and 0.05 mgmL^−1^ for IC_50_ of DPPH and ABTS free radicals, respectively.

Researchers have expanded their interest to investigate the protective impact of fucoxanthin on damage caused by oxidation in biological systems due to the positive results of its antioxidant property. Fucoxanthin was found to significantly reduce lipid peroxidation and/or ROS generation in HaCaT cells, ARPE-19 cells, and RAW 264.7 macrophage cells by inducing catalase activity. This is because fucoxanthin can recover oxidative stress as it repairs DNA damage and retrieves the morphological changes in the cells [26,108,109,110]. Similar findings were found in an in vivo study with fucoxanthin supplementation [111,112,113]. Meanwhile, fucoxanthin was reported to induce cell death in human skin melanoma (A375) cells primarily by ROS production, where its photo-excitation caused cell damage. Hypothetically, the excitation energies with energy levels greater than the singlet excited state of singlet oxygen exhibit photo-sensitising characteristics thus, the energy transfer from photo-excited fucoxanthin to singlet oxygen will cause the oxygen production [114]. This finding suggests that fucoxanthin can induce cell death in cancer cells by elevating ROS production while not affecting normal cells [115,116].

A study by Kong, et al. [117] has stated that the anti-inflammatory and antioxidant activity of fucoxanthin caused a decrease in diabetic symptoms and signs, improving spermatogenesis and male reproductive function by hindering the generation of reactive oxygen species (ROS), nitric oxide (NO), and lipid peroxidation while restoring mRNA expression (SOC-3 and GPR54) in the hypothalamus, and recovering the LH and testosterone hormones. Likewise, Chiang, Chen, Chang, Shih, Shieh, Wang and Hsia [109] demonstrated the protective mechanism of fucoxanthin in the human retina of diabetic retinopathy. It helped recover cell damage, reduce inflammation responses, and maintain the blood-retinal barrier’s integrity by lowering protein expression of apoptosis and cell adhesion, where it down and upregulates ICAM-1 and occluding expression, respectively. Aside from its capability to scavenge free radicals and quench singlet oxygen, fucoxanthin was reported to exert its antioxidant activity by modulating Nrf2/ARE, ERK/p38, PI3 K/Akt, and Sirt1 signalling pathways, as well as altered production or expression of ROS, glutathione (GSH), glutathione S-transferase (GST), catalases, HO-1, NQO1, and apoptosis-related protein [109,111,112,113,118,119,120,121].

### 5.2. Anti-Inflammatory Activity

Over-production of pro-inflammatory cytokines/mediators causes several inflammatory diseases such as obesity-related type 2 diabetes, insulin resistance, non-alcoholic steatohepatitis (NASH), asthma, arthritis, inflammatory bowel disease, atherosclerosis, and multiple sclerosis [111,122,123,124]. Multiple in vivo and in vitro studies have been done on the protective effects of fucoxanthin [108,110,123,124,125,126]. A study showed that fucoxanthin treatment inhibits prostaglandin (PGE_2_) and NO production in a concentration-dependent manner through downregulation of COX-2 and iNOS enzymes expression, respectively. It also prevents degradation of IκB-α phosphorylation, resulting in suppressed NF-κB activation, while the phosphorylation of ERK1/2, p38 and JNK MAPKs was dramatically reduced, causing lowered MAPKs in the macrophages of lipopolysaccharide-induced RAW 264.7 cells [125]. A similar reduction pattern was observed in TNF-α-stimulated HaCaT keratinocytes, LPS-stimulated THP-1 macrophages, and Hepa1–6 cells, where fucoxanthin treatment attenuated TNF-α, MCP-1, IL-1β, and IL-6 production [110,111].

In an animal study by Hosokawa, Miyashita, Nishikawa, Emi, Tsukui, Beppu, Okada and Miyashita [123], dietary fucoxanthin inhibited the expression levels of pro-inflammatory adipocytokines mRNA (TNF-α and MCP-1) associated to insulin resistance, resulting in over-inflammation of the liver. It also was found to significantly reduce mRNA expression levels (PAI-1 and IL-6) in the mesenteric WAT in diabetic/obese KK-A^y^ mice. Rodríguez-Luna, Ávila-Román, González-Rodríguez, Cózar, Rabasco, Motilva and Talero [110] demonstrated that fucoxanthin-supplement cream could prevent UVB-induced skin erythema and epidermal hyperplasia in mice with a substantial reduction of skin oedema shown when compared with the TPA-cream group. Myeloperoxidase (MPO) activity was also lowered by fucoxanthin cream, which reduced neutrophil infiltration relevant in the hyperplasia model. The UVB-induced erythema was reduced via the COX-2 and iNOS downregulation, and also HO-1 protein upregulation of through the Nrf-2 pathway.

Kawashima (2011) proposed that fucoxanthin and fucoxanthinol can improve inflammatory diseases by inhibiting Th17 cell development and stimulating Foxp3^+^ Treg cell differentiation as retinoic acid (RA) in a dose-dependent manner. Unlike fucoxanthin, other carotenoids such as astaxanthin, lutein, and lycopene did not affect T cell differentiation. In contrast, Hwang, Phan, Lu, Hieu and Lin [124] revealed that in Caco-2 cells, low-molecular-weight fucoidan (LMF) and high-stability fucoxanthin (HS-fucoxanthin) operate as anti-inflammatory agents and prebiotics where the combination of them significantly improved the intestinal immune function and epithelial barrier against the lipopolysaccharide effect via IL-1β and TNF-a inhibitions as well as IL-10 and IFN-γ promotions. Another study has proposed that a combination of rosmarinic acid and fucoxanthin has anti-inflammatory effects in UVB-exposed HaCaT keratinocytes as they downregulate inflammasome components ASC, caspase-1, NLRP3, and IL-1β, signifying that the combination exerts photo-protective effects to the constantly exposed skin [108].

### 5.3. Anti-Obesity Activity

Obesity is anticipated to affect more than a billion people by 2030, which is twice as many as in 2010 [127]. It is a global hazard to humans since it can lead to a variety of diseases such as metabolic syndrome, diabetes mellitus type 2 (DM2) and cardiovascular diseases (CVD). In animal studies, fucoxanthin results in weight reduction qualities through stimulation of mitochondrial uncoupling protein 1 (UCP1), which functions as metabolic thermogenesis to prevent excessive fat and lipid formation as well as inhibition of adipocyte differentiation. It can inhibit the glycerol-3-phosphate dehydrogenase action and downregulate peroxisome proliferator-activated receptor γ (PPARγ) which is important for the expression of the adipogenic gene [128,129].

Several studies reported fucoxanthin could attenuate body and WAT weight in C57 Bl/6 J and KK-A^y^ mice fed with a diet high in fat, which has no significant influence in food consumption when compared to groups without fucoxanthin supplementation [130,131,132,133]. Almost all WAT adipocytes were lesser in fucoxanthin-fed mice compared to the control mice, implying that fucoxanthin affects not only WAT size but also adipocyte characteristics in KK-Ay mice [123,130]. According to Gille, Stojnic, Derwenskus, Trautmann, Schmid-Staiger, Posten, Briviba, Palou, Bonet and Ribot [130], fucoxanthin can downregulate mRNA levels of lipolysis-related genes (Lipe and Plin1), fatty acid uptake-related gene (Cd36), lipogenesis-related genes (Srebf1), and lipoprotein lipase coding (Lpl) in mice visceral WAT. In a human clinical trial by Abidov, Ramazanov, Seifulla and Grachev [129], an intake of 2.4 mg/d of fucoxanthin for 16 weeks was reported to significantly decrease body weight, body fat, waist circumference, serum triglycerides level, plasma aminotransferase enzymes level, and blood pressure level while increasing resting energy expenditure (REE). Stimulation of REE was partly responsible for the loss of body weight and body fat reduction, combined with wide anti-inflammatory and metabolism-normalising activities. It was proposed that pomegranate oil (PO) could assist in promoting the REE-stimulating action of fucoxanthin, where an increase of REE might be connected to fucoxanthin’s activation of UCP proteins, as formerly demonstrated by Maeda, et al. [134] and Nicholls and Locke [135]. On the contrary, Mikami, et al. [136] reported that fucoxanthin alone without the combination of PO did not change body weight, body mass index (BMI), and REE activity in normal-weight and obese subjects. This could be presented by the significant differences among the research population (Russian and Japanese), as well as the presence of PO.

Mitochondrial UCP1 usually appears in brown adipose tissue (BAT) and minute amounts in WAT. Unlike WAT, BAT is less found in adult humans. Thus, the presence of UCP1 in WAT theoretically can reduce abdominal fat [134]. Several studies showed that the administration of fucoxanthin from various species of brown seaweeds elevated the UCP1 expression in the WAT of db/db mice and diabetic/obese KK-Ay mice fed with a high-fat diet, resulting in β-oxidation promotion and attenuation of the leptin expression and increase adiponectin serum levels [133,137,138,139]. According to Kim, et al. [140], the intervention of 3 mg fucoxanthin for 3 months initiated BAT expression in healthy obese individuals, where BAT was found in supraclavicular, cervical, and paravertebral spaces. Though little to no weight or laboratory changes were observed in the study, a parallel effect was observed in an animal study by Gille, Stojnic, Derwenskus, Trautmann, Schmid-Staiger, Posten, Briviba, Palou, Bonet and Ribot [130].

Okada, Mizuno, Sibayama, Hosokawa and Miyashita [133] demonstrated that diets of *Undaria pinnatifida* lipids (UL) containing fucoxanthin and its metabolites combined with scallop phospholipids (PL) capsules; high in n-3 polyunsaturated fatty acids caused a substantially increase the UCP1 expression (2.15-fold higher compared to the control group), indicating that UL and PL work synergistically to reduce obesity and exhibit greater anti-obesity effect. On the other hand, Mikami, Hosokawa, Miyashita, Sohma, Ito and Kokai [136] revealed that 2 mg/d of fucoxanthin intake in a clinical trial of 60 Japanese adult men and women of normal and obese weight markedly reduced HbA_1 c_ and glycated albumin levels in subjects possessing the UCP1–3826 A/G thrifty allele when compared to those with the A/A and A/G genotype. Since UCP1 G/G genotype is a well-studied gene linked to the onset of obesity and insulin resistance, it is possible to cater specific treatment based on what is needed via SNP analysis as this G/G allele individuals might be at higher risk of developing the obesity-related disorder.

High expression of adipocytokine production in WAT acts as one of the obesity markers. Fucoxanthin was reported to markedly reduce mRNA adipocytokines expression in diabetic/obese KK-Ay mice [123,132]. Palmitic acid (PA) (saturated fatty acids) is discharged from adipocytes; increased secretion of TNF-α from macrophages can further induce lipid accumulation [141]. High stability fucoxanthin (HS-fucoxanthin) is known to reduce oxidative stress caused by palmitic acid without impacting cell viability while limiting PA-induced lipid accumulation at high concentration via long non-coding RNA (lncRNA) modulation, HULC, HOXA-3 AS, IPW, SCA-8, PCAT-43, and PCAT-29 in PA-BSA-induced adipocyte hypertrophy [123,141]. On the contrary, the anti-obesity effect of fucoxanthin and its metabolites was assumed to be due to its regulatory influence on adipocyte differentiation. Dose-dependent fucoxanthin could enhance early-stage (D0-D2) 3 T3-L1 adipocyte differentiation and induce lipid accumulation via upregulation of the key transcriptional regulators (PPARγ, C/EBPα, and SREBP1 c) expression, transcription factors; regulated the adipogenic gene expression as well as a marker of adipocyte differentiation (aP2). However, fucoxanthin reduces PPAR, C/EBP, and SREBP1 c levels at D2-D4, D4-D7, and D2-D7 during intermediate and late-stage differentiation, finally resulting in restricted glucose absorption in mature 3 T3-L1 adipocytes through inhibiting IRS-1 phosphorylation [142]. Comparable results were obtained in a study by Lai, et al. [143] involving 10 μg/mL of xanthigen (pomegranate seed oil mixed with brown marine algae fucoxanthin) on additional modulation of SIRT-1 protein expression, AMPK, and FoxO pathways.

Finally, although few studies have been reported, the inhibitory effect of fucoxanthin and its metabolite, fucoxanthinol on rat pancreatic lipases has been investigated. Both products were reported can inhibit triolein hydrolysis with IC_50_ of 660 and 764 nM, respectively, which are nearly 100-fold higher than orlistat IC_50_ (6.8 nM). Moreover, fucoxanthin or fucoxanthinol has been demonstrated to reduce lymphatic lipid absorption and lower triglyceride concentrations in the systemic circulation. Eventually, the amount of free fatty acids and monoacylglycerols in the intestinal lumen reduces when pancreatic lipase is inhibited [90]. Howbeit, Guo, Oliviero, Fogliano, Ma, Chen and Capuano [97] reported that pancreatic lipase majorly contributes to the deacetylation of fucoxanthin to fucoxanthinol due to its esterase activity where approximately 53% of initial fucoxanthin was detected as fucoxanthinol after digestion with pancreatin.

### 5.4. Antidiabetic Activity

According to the World Health Organisation (WHO), diabetes affects over 422 million people globally and the number is expected to increase by 25% in 2030 and 51% in 2045 [144]. To date, there is no cure for diabetes. However, preventive measures have been taken to reduce the complications and mortality rate caused by this disease, including early diagnosis and healthy lifestyle practices. An active compound that can help induce insulin intake and reduce blood sugar levels in blood remained to be the goal in diabetes treatment along with reducing diabetes complications such as heart attacks, kidney failure, limb amputation, blindness, and stroke [145,146]. Obesity is directly correlated with DM2, where inflammatory responses can lead to insulin resistance or might be amplified by the hyperglycaemic state, resulting in DM2 complications [147]. Dysregulation of pro-inflammatory adipocytokines production in WAT such as plasma leptin, tumour necrosis factor-a (TNF-α), and non-esterified fatty acid concentrations with the downregulation of anti-inflammatory adiponectin can lead to infiltration of macrophages into WAT causes chronic, low-grade inflammation that closely related to insulin resistance and obesity- DM2 [123,148].

MCP-1 is released into the bloodstream by abdominal WAT, limits insulin-dependent glucose absorption and contributes to insulin resistance along with the increased level of blood TNF-α and IL-6 concentration [88]. In contrast, dietary supplementation of fucoxanthin and/or low-molecular-weight fucoidan substantially reduces concentrations of plasma insulin, blood glucose, blood HbA_1C_, and resistin levels in high-fat diet-fed C57 BL/6 N mice, diabetic/obese KK-Ay mice, and DM2 db/db mice compared to the control group [130,132,149,150,151]. This might happen because fucoxanthin could inhibit macrophage infiltration in both perigonadal in KK-Ay mice and mesenteric WAT in diabetic/obese KK-Ay mice. MCP-1 and TNF-α mRNA expression is significantly decreased, supporting the decrease of blood glucose levels while fucoxanthinol markedly inhibited the MCP-1 mRNA overexpression (TNF-α-induced) in 3 T3-F442 A cells. Nonetheless, fucoxanthin did not influence the blood glucose level in normal feed C57 BL/6 J mice [123,132]. Overexpression of IL-6 mRNA and IL-6 generation was also attenuated by fucoxanthin in 3 T3- F442 A (TNF-α-stimulated) cells, as well as in mesenteric and perigonadal WAT of both KK-Ay mice and diabetic/obese KK-Ay mice [123]. The plasminogen activator inhibitor-1 (PAI-1) level is higher in DM2 and metabolic syndrome, whereby nature fucoxanthin functions as a thrombosis and fibrosis mediator, which explains its significant risk factor for macrovascular complications and cardiovascular diseases, particularly in patients with diabetes [152]. On the other hand, intake of fucoxanthin tends to lessen expression levels of PAI-1 mRNA in perigonadal WAT of KK-Ay mice while substantially reducing mesenteric WAT of diabetic/obese KK-Aγ and markedly decreasing in co-culture cells of 3 T3-L1 adipocyte and RAW264.7 macrophage cells [123,132].

Furthermore, glucose transporter 4 (GLUT4) is acutely induced by insulin and remarkably expressed in adipose tissue and skeletal muscle to promote glucose uptake into the body tissues. However, the translocation of GLUT 4 into the plasma membrane in an insulin resistance scenario is not favourable [153]. Nevertheless, Nishikawa, Hosokawa and Miyashita [151] reported that fucoxanthin treatment on KK-Ay mice gave a notable elevation in translocation of GLUT4 to plasma membranes in the soleus muscle and improved EDL’s muscle translocation. Hence, GLUT4 expression was significantly increased in the EDL muscle at once, but not in the soleus muscle. These findings indicate that translocation and expression of GLUT 4 efficiency exerted by fucoxanthin are different depending on the muscle types. In an insulin-resistant subject, the insulin pathway is disrupted, where the activation of IR tyrosine kinase is inhibited, resulting in no phosphorylation of Akt/protein kinase B via the tyrosine site phosphorylation of IRS-1 and further attenuating the translocation of GLUT4 into the plasma membrane. However, fucoxanthin can reverse this situation by markedly increasing the expression of IR mRNA by activating phosphorylation of Akt in both muscles compared to the control group. Fucoxanthin also upregulated PGC-1 α expression levels in both muscles, similar to the EDL muscles, 190% higher than the control group [151].

Research by Lin, Tsou, Chen, Lu and Hwang [149] revealed that supplementation of low-molecular-weight fucoidan (LMF) and/or fucoxanthin with high stability helps in stimulating serum adiponectin levels while slightly decreasing serum insulin levels in comparison to the control. However, this finding contradicts Hosokawa, Miyashita, Nishikawa, Emi, Tsukui, Beppu, Okada and Miyashita [123], where no changes were recorded in serum adiponectin levels; stated that fucoxanthin’s effects on WAT weight and blood glucose levels are not due to adiponectin synthesis, but rather to another mechanism. The study was further extended by investigating glycosuria events in db/db mice, where they observed LMF and fucoxanthin supplementation slightly reduced the level of urinary glucose. A substantial reduction was detected in LMF + fucoxanthin, demonstrating that combined treatments exerted a synergistic effect. The study also showed that db/db mice had higher mRNA expression of the transcription factor peroxisome proliferator-activated receptor (PPAR), where the activation of PPARγ inhibited TNF-α expression in the obese rodents’ adipose tissue.

Carbohydrate metabolism enzymes, such as α-amylase hydrolyses oligosaccharides and α-glucosidase, will further metabolise the disaccharides into monosaccharides for intestinal absorption. Inhibition of the enzyme’s activities can control diabetes by reducing glucose absorption [154]. Intriguingly, fucoxanthin is reported to inhibit the action of these enzymes, and these discoveries suggest that fucoxanthin can be beneficial as a functional dietary component for glycaemic control. Kawee-Ai, et al. [155] showed that fucoxanthin demonstrated high inhibition of α-amylase in a concentration-dependent manner, with an IC_50_ value of 76.92 μg/mL, but weak inhibition of α-glucosidase, having an IC_50_ value of 537.32 μg/mL. Another study has reported a low inhibitory effect of fucoxanthin in *Undaria pinnatifida* extract against α-glucosidase with an IC_50_ value of 80 μg/mL where a higher IC_50_ inhibition value with fucoxanthin analytical standard of 47 μg/mL was observed [156]. Meanwhile, fucoxanthin in *Iyengaria stellata* (IC_50_ 0.33 μg/mL) and *Colpomenia sinuosa* (IC_50_ 3.50 μg/mL) extract was reported have stronger inhibitory effect on α-glucosidase compared to acarbose inhibition (IC_50_ 160.15 μg/mL) [157]. Nevertheless, Wang, et al. [158] have reported that fucoxanthin administration can improve alpha-glucosidase activity in cisplatin-induced hamsters after treatment for 5 days.

### 5.5. Anticancer Activity

Cancer develops when abnormal cells undergo uncontrolled division and metastasise to the surrounding tissues. As of 2020, WHO reported that cancer is the second largest cause of mortality worldwide, as 10 million people died of cancer, and it is 2.5 times more common in wealthy countries [159]. Cancer is identified by six features: evading growth suppressor, maintaining proliferative signal, evading defying growth suppressor, allowing replicative immortality, activating invasion and metastasis, promoting angiogenesis, and preventing cell death [160]. Fucoxanthin and fucoxanthinol are reported as potential potent therapeutic agents in cancer treatment, as they can induce cell cycle arrest, reduce cell viability and cancer cell apoptosis, and suppress cancer cell metastasis and anti-angiogenesis effect.

Cell proliferation is vital in the process of tumour or cancer formation. Cytotoxicity of fucoxanthin diminishes when administered below 40 μM concentration [161,162]. Meanwhile, 150 μM of fucoxanthin was cytotoxic towards non-tumorigenic cells (40%) compared to control cells [163]. Thus, administration of fucoxanthin and/or fucoxanthinol in a dose- and/or time-dependent manner is significant in decreasing many cancer cells’ viability such as glioblastoma cell lines (GBM1, A172, C6, U87 and U251) [163,164], leukaemia cell lines (RAW264.7) [112], human colorectal cancer cells (HT-29, HCT116, and DLD-1 cells) [165,166], human lymphatic endothelial cells [167], lung carcinoma (A549), lung cancer (NCI-H522), colon adenocarcinoma (WiDr and Lovo), breast adenocarcinoma (MCF), neuroblastoma (SK-N-SH), hepatocellular carcinoma (Hep G2), malignant melanoma (Malme-3 M), cervix squamous (SiHa) [168], EBV-immortalised human B-cell lines, BL, and HL cell lines [169], human osteosarcoma cell lines (Saos-2, MNNG/ HOS [MNNG] and 143 B) and mouse osteosarcoma cell lines [170], human bladder cancer cells (T24) [171], and mice xenografted sarcomas 180 [172]. According to Rokkaku, Kimura, Ishikawa, Yasumoto, Senba, Kanaya and Mori [170], β-carotene and astaxanthin showed an insignificant effect on cell viability reduction, while fucoxanthin and fucoxanthinol reduced cell viability significantly in all four osteosarcoma cell lines in a dose-dependent manner. A previous study by Kotake-Nara, et al. [173] evaluated 15 carotenoids found in foods on the development of human prostate cancer cell lines and reported that among all carotenoids, neoxanthin and fucoxanthin (allenic carotenoids) exhibited the highest activity, decreasing cell viability more than those without allenic bond, which signals that the allenic bond is the main element for anti-proliferative ability of carotenoids.

Fucoxanthin intake has been reported to suppress the amount and growth of tumours in animal models [174]. Numerous studies have targeted biomolecules and signalling pathways associated with the cell cycle arrest, anti-metastasis, apoptosis, and angiogenesis suppression to examine the underlying mechanisms of fucoxanthin’s anticancer potential [171,175,176,177,178]. These studies indicate that fucoxanthin can suppress the cell cycle in G0/G1, S, and/or G2/M phase depending on the cancer cell types by modulating several genes and protein expression, involving Mcl-1, STAT 3, p-STAT3, survivin, Bcl-2, Bcl-x, cIAP-2, XIAP, c-Myc, cyclin-dependent kinases (CDKs), and cyclins [169,170,171,177,179]. Apoptosis, or the death of cancer cells, is the best way to stop and treat cancer. Through targeting various molecular paths, fucoxanthin can induce apoptosis by altering several pathways, including the JAK/STAT signalling pathway, PI3 K/Akt/NF-κB signalling, and abruption of mortalin–p53 complex, and caspase activation [167,169,171,174,177,178,179,180]. Inhibitors of angiogenesis are an essential aspect of cancer prevention. Ganesan, Matsubara, Sugawara and Hirata [176] demonstrated that fucoxanthin suppressed fibroblast growth factor 2 (FGF-2) mRNA expression, its receptor (FGFR-1), and its trans-activation factor (EGR-1). Then, it further downregulates the phosphorylation of FGF-2-mediated intracellular signalling proteins (ERK1/2 and Akt), which eventually deterred endothelial cells movement and differentiation into tube-like formations on the Matrigel. Other anti-angiogenesis effects of fucoxanthin are reported in additional studies [170,179,181].

Additionally, Terasaki, Maeda, Miyashita and Mutoh [166] investigated the induction of anoikis, as well as anchorage-dependent apoptosis in colorectal cancer (CRC) by fucoxanthinol. Their results indicate that exposure to 2.5 µM fucoxanthinol showed apoptotic and anti-proliferative effects on DLD-1 cells. However, during 6 to 72 h of fucoxanthinol treatment, an increasing trend in the number of floating living cells was observed compared to the non-treatment group. Later, a Western blot analysis revealed that after 6 h of treatment, fucoxanthinol made FAK less active and changed how integrin 1 was expressed and where it was found. After 24 h, the cells’ expression of PPAR and activation of Akt decrease, while their expression of integrin 1 increases. The results showed that fucoxanthinol can kill CRC cells by stopping the signals from integrins in human CRC cells. The high polarity of fucoxanthin and fucoxanthinol compared to low- or non-polar carotenoids caused it to be rapidly absorbed into cancer cell organelles, resulting in decreased mitochondrial membrane potential and activation of integrin signal. Disorder in the potential of the mitochondrial membrane leads to the release of apoptosis’s trigger molecules (Endo G, AIF and cytochrome c) out of the mitochondria into the cytosol.

### 5.6. Other Activities

Fucoxanthin has a neuroprotective behaviour against oxidative damage caused by hydrogen peroxide (H_2_O_2_) [180,181,182,183] and β-amyloid oligomers [149,184]. This was believed due to the PI3 K/Akt cascade activation and ERK pathway inhibition by fucoxanthin [185,186]. The accumulation of Aβ plaques caused a major neurotoxin in Alzheimer’s disease (AD), and fucoxanthin exhibited a potent effect in reducing the formation of Aβ plaques in both animal and cell models [182,184]. It is common to see a neurodegenerative pattern caused by oxidant damage and microglial activation and inflammation in AD patients. Fucoxanthin demonstrated a promising effect where it can ameliorate inflammation as well as oxidative stress in BV2 microglia cells via suppressed MAPK phosphorylation pathways, causing reduced pro-inflammatory secretion in BV2 and recovery of antioxidative enzymes (superoxide dismutase; SOD and glutathione; GSH), resulting in a significant reduction of intracellular ROS [183,185]. On the other hand, fucoxanthin treatment was reported to fix the memory loss caused by scopolamine in the novel object recognition (NOR) test in an animal model. It was observed that fucoxanthin extensively reversed the rise of acetylcholinesterase (AChE) activity and the reduction of choline acetyltransferase (ChAT) activity in the hippocampus and cortex caused by scopolamine, leading to an increased brain-derived neurotrophic factor (BDNF) expression, where AChE is directly restrained by fucoxanthin, which has an IC_50_ value of 81.2 μM. in a non-competitive manner [187]. Comparable BDNF production was also reported by Zhao, Kwon, Chun, Gu and Yang [185]. Additionally, fucoxanthin can ameliorate traumatic brain injury and cerebral ischemic/reperfusion injury in animal models by stimulating Nrf2- ARE and Nrf2-autophagy pathways and Nrf2/HO-1 signalling [113,188]. The β-site amyloid precursor protein cleaving enzyme 1 (BACE1) action is closely linked to the AD onset. Fucoxanthin exhibited mixed-type inhibition against BACE1, where two of its hydroxyl groups interact with BACE1 residues, Gly11 and Ala127, making it a potential treatment compound in treating AD [189]. Moreover, Paudel, et al. [190] believed that fucoxanthin might be valuable in managing neurodegenerative diseases, particularly Parkinson’s disease, as it showed a potential dopamine D3/D4 agonist.

Apart from that, recent studies revealed that fucoxanthin treatment showed antifibrotic activity. Fucoxanthin demonstrated a potential protective behaviour on pulmonary fibrosis and nasal polyps, where it attenuates the expression/production of α-smooth muscle actin (α-SMA), type 1 collagen (Col-1), fibronectin, and IL-6 in transforming growth factor-beta1 (TGF-β1)-stimulated cells via suppressed MAPK phosphorylation, PI3 K/Akt pathway, Akt/SP-1 pathway, and Smad2/Smad3 pathway [191,192]. Fucoxanthin is also reported to exert a bacteriostatic effect and suppressed arylamine-N-acetyltransferase (TBNAT) and UDP-galactopyranose mutase (UGM) in the study of antitubercular properties of fucoxanthin [193]. The relatable antibacterial effect of fucoxanthin has been documented in another report by [194]. Excess osteoclast activity caused an imbalanced bone remodelling, favouring resorption and induced osteoporosis, rheumatoid arthritis, periodontal disease, metastatic cancers and multiple myeloma [195]. Fucoxanthin further showed a potent suppressive effect on osteoclastogenesis by inhibiting osteoclast differentiation and initiating apoptosis in osteoclasts, while no cytotoxicity was observed against osteoblasts [195,196]. However, to date, fucoxanthin treatment does not support the idea as a potential therapeutic effect for bone repair because only a few studies support this perception while others showed insignificant results [197].

Antiapoptotic and antioxidant properties of fucoxanthin showed a protective impact on kidney cells. To date, chronic kidney disease (CKD) has no optimal clinical treatment in hindering and treating it from becoming worse. Antiapoptotic properties of fucoxanthin showed a protective impact by upregulated Na + /H + exchanger isoform 1 (NHE1) expression in rat renal tubules but not in PPARα knockout mice. The antiapoptotic activity of fucoxanthin reportedly reduces renal tubulointerstitial fibrosis and improves renal function in CKD mice [198]. Combination treatment of oligo-fucoidan, fucoxanthin, and L-carnitine showed a great impact on renal function and weight as they inhibited renal fibrosis, reduced serum creatinine level, activated Akt, and inhibited H_2_ O_2_-induced apoptosis in rat renal tubular cells [199]. The antioxidant property of fucoxanthin ameliorates fibrosis and oxidative stress induced by high glucose (HG) via Akt/Sirt1/FoxO3α signalling in glomerular mesangial cells, indicating its idealism in the treatment of diabetic nephropathy (DN) [200]. Recently, fucoxanthin regulates early transcriptomic and epigenomic biomarkers along the course of HG-induced DN, assisting mesangial cells in their protection against HG-induced oxidative damage [201].

Excess lipid accumulation in hepatocytes can cause liver disease. Fucoxanthin can keep triglycerides, cholesterol esters and total cholesterol from accumulating in the liver via suppressed mRNA expression of lipogenesis-related genes, cholesterol esterification, lipid droplet accumulation, and induced CPT1 A mRNA level (β-oxidation related gene) [202]. Fucoxanthin also elevates levels of non- and HDL-cholesterol in KK-Aγ mice by generating SREBP expression and reducing the liver’s cholesterol uptake through downregulation of SR-B1 and LDLR, resulting in raised mice’s serum cholesterol [203]. Besides that, fucoxanthin can accelerate the promotion of omega-6 PUFA and omega-3 PUFA to arachidonic acid (AA) and docosahexaenoic acid (DHA), eventually causing an improvement in lipid profile [89]. A recent study by Takatani, Kono, Beppu, Okamatsu-Ogura, Yamano, Miyashita and Hosokawa [111] revealed that supplementation of fucoxanthin could substantially reduce liver weight gain, hepatic lipid oxidation, hepatic fat accumulation and mRNA expression levels of inflammation, as well as infiltration-related genes. Furthermore, the metabolites of amarouciaxanthin A and fucoxanthinol showed anti-inflammatory properties through suppressed chemokine production in hepatocytes. Fucoxanthin is also thought to reduce cardiovascular disease-related risk factors by modulating ACE, α-amylase, and α-glucosidase activities [40]. Overall, the mechanism of fucoxanthin resulted in several biological activities that are beneficial to human health, which are further summarised in Table 2.

## 6. Fucoxanthin as Functional Food Ingredients

Fucoxanthin is known for its therapeutic activities that can benefit humans. It can be incorporated into several products, especially food, cosmetics, and pharmaceuticals. Fucoxanthin is generally halal, kosher, as well as vegetarian and vegan friendly. Other seaweed extracts such as carrageenan, alginate and fucoidan (macromolecules) as well as astaxanthin and β-carotene (micromolecules) are already acknowledged by consumers worldwide [207]. Generally, natural functional seaweed extract is available as a supplement for retail consumers at health stores and online. It is commercialised as dried seaweed extract, commonly from kombu or wakame, as expensive weight loss supplements dependent on its qualities and purity, ranging from 10% to 98%. According to Miyashita and Hosokawa [208], fucoxanthin can be incorporated as a food ingredient to hasten adaptive thermogenesis. The European Food Safety Authority (EFSA) publicly consented to scientific statements on fucoxanthin consumption and its health claims verification as a body weight loss supplement in 2009. It was indicated that fucoxanthin extract of *U. pinnatifida thallus* can be sufficiently consumed at an amount approximate to 15 mg per day (Article 13(1), Regulation (EC) No 1924/2006). Nevertheless, as yet the EFSA has not approved the relationship between fucoxanthin consumption and the normal body weight accomplishment [209]. Therefore, it is generally sold as a common food supplement in countries such as the United States, Japan, Ireland and many more as oil form or microencapsulated powder. It is not sold as pure fucoxanthin, as the extraction cost of bulk food ingredients is high, and its instability to oxidation has been prohibitive. Fucoxanthin is available as various seaweed extracts concentration, where analytical grade of fucoxanthin can be obtained at the purity of ≥95% for food producers and laboratory usage [21].

Fucoxanthin extract may encounter some challenges to be incorporated into food from several aspects such as chemical, organoleptic, bioavailability, and stability. The pure form of fucoxanthin is not stable enough as it is easily oxidised by a few factors such as pH, temperature, and UV light [89,206]. Theoretically, the chemical structure of fucoxanthin may have interactions with other food ingredients during preparation or storage. For example, bakery food products require high temperature, while beverages such as juice, yogurt, and honey may have low pH, which eventually causes deterioration of fucoxanthin content. As fucoxanthin originates from aquatic plants, the organoleptic attributes such as appearance, texture, smell, and taste need to be considered a major factor affecting food quality. Fucoxanthin has a natural brown colour that may affect the food’s appearance. Moreover, the savoury taste and fishy smell of seaweed might potentially be carried over into fucoxanthin extract, rendering it unpalatable to some people [21]. The chance of fucoxanthin bioavailability being affected during its metabolism in the human body is possible. It is insoluble in water, and the presence of dietary lipid is required to create emulsification or colloids dispersion for better solubility and adsorption [16,210,211,212,213,214,215,216,217]. Koo, et al. [210] reported that fucoxanthin was not stable n in water as fucoxanthin content in water degenerated approximately 30% of the original amount at the storage of 26 °C after 4 weeks.

Nonetheless, Oryza Oil and Fat Chemical Co., Ltd. [211] has developed oil and powder products of fucoxanthin from kombu (*Laminaria japonica*) having 1–5% fucoxanthin. The constituents of the product are kombu extract, natural tocopherol (stabilising factor), and cyclodextrin and triglyceride (which provide a protective matrix to the fucoxanthin). The fucoxanthin fraction of the product was stable up to 80 °C (1 h) at pH 3–10, with a maximum loss of 6% (pH 3) after 1 week. The powder and oil products of fucoxanthin have been effectively integrated into various foods such as beverages, cakes, shortbread, spreads, potato snacks, etc. In the meantime, several studies have incorporated fucoxanthin as functional food ingredients, as summarised in Table 3. 

Prabhasankar, Ganesan, Bhaskar, Hirose, Stephen, Gowda, Hosokawa and Miyashita [218] have investigated the incorporation of *U. pinnatifida* (wakame) powder, containing fucoxanthin and fucosterol, into semolina wheat-based pasta. The study showed that the powder could influence the sensory, biofunctional, and nutritional qualities of pasta. The sensory test found no significant result on organoleptic (appearance, mouth-feel, taste, and strand quality) discrepancies between the wakame pasta and the control. However, the acceptance decreased at more than 10% wakame powder concentration, and when it reached 20%, the sensory attribute was affected by the saltiness and seaweed taste. Throughout pasta preparation, an approximate loss below than 10% for fucoxanthin and fucosterol was detected after the kneading and cooking processes due to the stability of both compounds in the gluten protein matrix, i.e., about 20% enhancement of interaction between protein matrix and starch granules in pasta comprising seaweeds as ingredients were reported.

Meanwhile, food-grade lipid extract containing fucoxanthin (2.75 g/100 g) was added to plain yogurt at 0.25% and 0.5% *w*/*v* and was investigated for more than 28 days of storage to evaluate product suitability. The study has shown no significant influence of the sample on yogurt composition, pH, viscosity, whey separation, or starter culture viability [214]. However, sensory analysis of both yogurts perceived negative sensory attributes compared to the control yogurt. A recent study by Zahrah, Amin and Alamsjah [212] has incorporated fucoxanthin as a colouring agent for shrimp paste to improve the appearance and also bioactive and antioxidant activities of the sample. According to the finding, the higher the concentration of fucoxanthin dye applied to the sample, the higher the Hue value (colour attribute referring to the lightness or darkness). Fucoxanthin pigment can provide colour due to the existence of chromophores and ausochromes [220], which react with amino acids in shrimp paste to produce colouring agents. Generally, the fucoxanthin concentration added to shrimp paste can improve colour but does not significantly affect the sensory attributes such as texture, taste, and odour of shrimp paste, but the study concluded that 12% fucoxanthin was the optimal concentration.

Moreover, O’Sullivan [216] has developed yogurt and fluid milk using *Fucus vesiculosus* and *Ascophyllum nodosum* methanolic extract that contained fucoxanthin. The milk’s overall acceptance was governed by the presence of fishy flavour. In contrast, the appearance (less green or yellow colour) and flavour of the yogurt were acceptable when incorporated with 100% water extract of *A. nodosum* at a 0.5% addition level. Furthermore, no deterioration in antioxidant bioactivities was noticed during the shelf-life study and pH conditions in the seaweed-supplemented milk and yogurt formulations. Meanwhile, Kartikaningsih, Mufti and Nurhanief [213] produced dried tea *Sargassum cristaefolium* containing fucoxanthin pigment at 1.64%. Fucoxanthin in dried tea was observed to have an orange/yellow colour when prepared at pH 6 and 2. However, when tea was prepared at pH 2, no allenic group was detected by the Fourier-transform infrared (FTIR) spectrophotometer in the sample, and the fucoxanthin solution also changed to a pale-yellow colour. Theoretically, fucoxanthin without an allenic bond is not active [88]. Thus, the study concluded that fucoxanthin is not stable in acidic solutions. Wang, et al. [221] reported that the allenic bond is unstable, weak, and easily breaks down. Similarly, Yip, Joe, Mustapha, Maskat and Said [41] also reported that fucoxanthin extracted from *Sargassum binderi* is only stable at pH 5 to 8. The fucoxanthin can be maintained by storing at low temperatures, reduced light exposure, and in non-metal containers because it is susceptible to deterioration in the food matrix, which contains transition metals, and it is more stable in a milk protein-containing food matrix [215,222,223].

Mok, Yoon, Pan and Kim [215] investigated the fucoxanthin content in fucoxanthin-fortified milk throughout storage, pasteurisation, and drying operations under diverse conditions. Milk deterioration rate was observed to be positively associated with storage temperature throughout one month of storage. Fucoxanthin-fortified skimmed milk with a higher protein content performed better than whole milk during storage and drying. Milk proteins, such as casein and whey protein isolates, have been reported as good encapsulation materials because they can stabilise materials by scavenging free radicals of sulfhydryl and non-sulfhydryl amino acids [224]. Approximately 91% of fucoxanthin content was preserved even after three pasteurisation processes. Nuñez de González, et al. [225] also studied the effect of fucoxanthin-fortified milk in pasteurised skim and whole goat milk. Both types of milk were observed to have high recovery yields of fucoxanthin after pasteurisation at 64 °C for 30 min (~96%) and were stable throughout the 4 weeks storage period (4 °C). Different fucoxanthin concentrations influenced the milk colour by increasing the yellowness. However, it does not affect milk composition or physicochemical properties.

Another fucoxanthin-based food assessment was done by Sugimura, Suda, Sho, Takahashi, Sashima, Abe, Hosokawa and Miyashita [217] in the bakery product of scones added with 0.5–2 wt% commercial dried wakame powder. The heating process (190 °C for 30 min) of dough slightly affected fucoxanthin content, where more than 85% of the fucoxanthin stayed stable in the food. The sensory analysis showed that the addition of wakame powder (0.5 wt% and 2 wt%) has darkened the appearance of the scones with better sensory acceptance (i.e., taste and smell) compared to the control. Next, the effect of fucoxanthin added to ground chicken breast meat was investigated by Sasaki, Ishihara, Oyamada, Sato, Fukushi, Arakane, Motoyama, Yamazaki and Mitsumoto [219], who reported that fucoxanthin could improve the appearance and shelf life of the meat. About 200 mg/kg of fucoxanthin was added to the sample and effects of chilling storage tests were performed before and after cooking. The thiobarbituric acid reactive substances (TBARS) test revealed insignificant results (approximately 3 nmol/g sample) for both raw meat; control and fucoxanthin-supplemented meat. Similarly, the cooked meat also gave insignificant TBARS value for control and fucoxanthin-supplemented meat of 58.1 and 34 nmol/g samples. In the meantime, TBARS result in fucoxanthin-supplemented meat (approximately 18 nmol/g) was significantly different with control (approximately 28 nmol/g) during 1-day chilling storage after cooking. Meanwhile, after 6 days of chilled storage after cooking, the TBARS value increased in both samples, but the increment in fucoxanthin-supplemented meat sample was lower (approximately 30 nmol/g) compared to the control (approximately 60 nmol/g). Overall, the colour indicator showed insignificant data for both storage periods.

## 7. Conclusions and Future Prospects

Brown algae is one of the established sources of fucoxanthin, a photosynthetic component and a very valuable aquatic carotenoid. Brown seaweed is a major source of fucoxanthin, as it may utilise the neoxanthin cycle or diadinoxanthin cycle for its formation. Brown seaweed is a potent source of fucoxanthin, as it is relatively more accessible, economical, and lower in value than microalgae. Metabolism of fucoxanthin generates the fucoxanthinol and amarouciaxanthin A metabolites that exhibit better bioavailability in the human body. Although it is chemically metabolised, the bioaccessibility of fucoxanthin is quite complex because of its non-polar, hydrophobic, and water-insoluble properties. Thus, fucoxanthin requires emulsification and colloid dispersion to help solubility and adsorption capability. Apart from that, fucoxanthin showed a promising impact in therapeutic activities. The combination of fucoxanthin with other compounds shows a synergistic effect with significant positive results. A number of in vitro and in vivo research have been carried out. However, to date, only a few human subjects have been involved, which could be further extended in the future, as the biological system between cells, rodents, and humans might differ and cause differential data. The direct addition or incorporation of fucoxanthin into food was intensively developed for the consumption of this nutraceutical. Past studies indicate that food quality formulated with fucoxanthin from algae is significantly influenced by several aspects, including chemical, organoleptic, bioavailability, and stability. It is concluded that the exploration and utilisation of fucoxanthin-derived functional foods can be marketed. Fucoxanthin, such as lutein and β-carotene, is a promising carotenoid and it should receive the attention it deserves in the food and pharmaceutical industries. Given the information presented in this review, fucoxanthin’s potential as a potent functional aquatic compound could be further explored to unravel any challenges and opportunities related to its application for human health.

## Figures and Tables

**Figure 1 foods-11-02235-f001:**
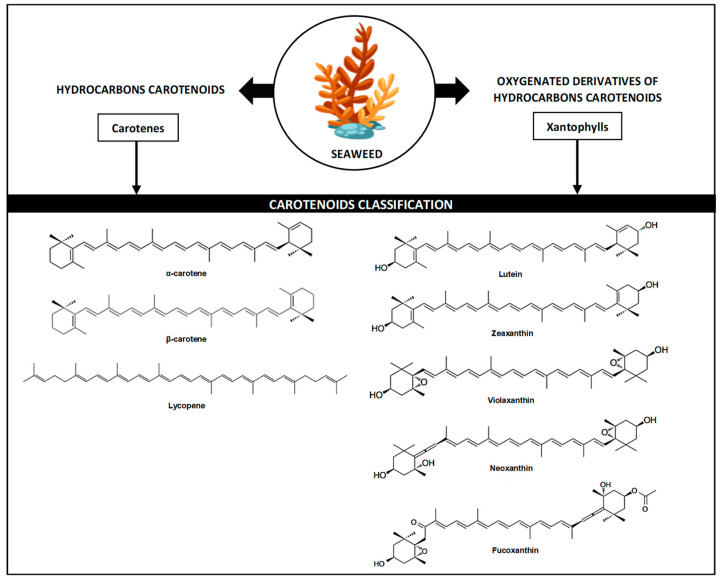
Summary of carotenoids classification in seaweed.

**Figure 2 foods-11-02235-f002:**
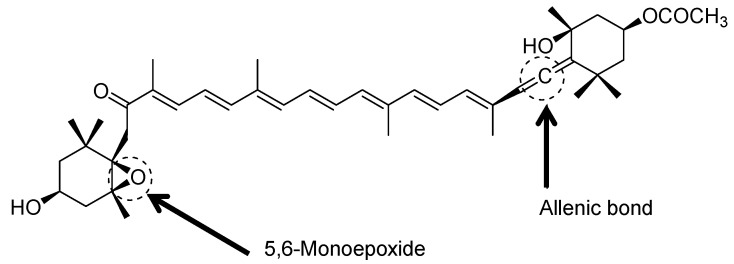
Chemical structure of fucoxanthin.

**Figure 3 foods-11-02235-f003:**
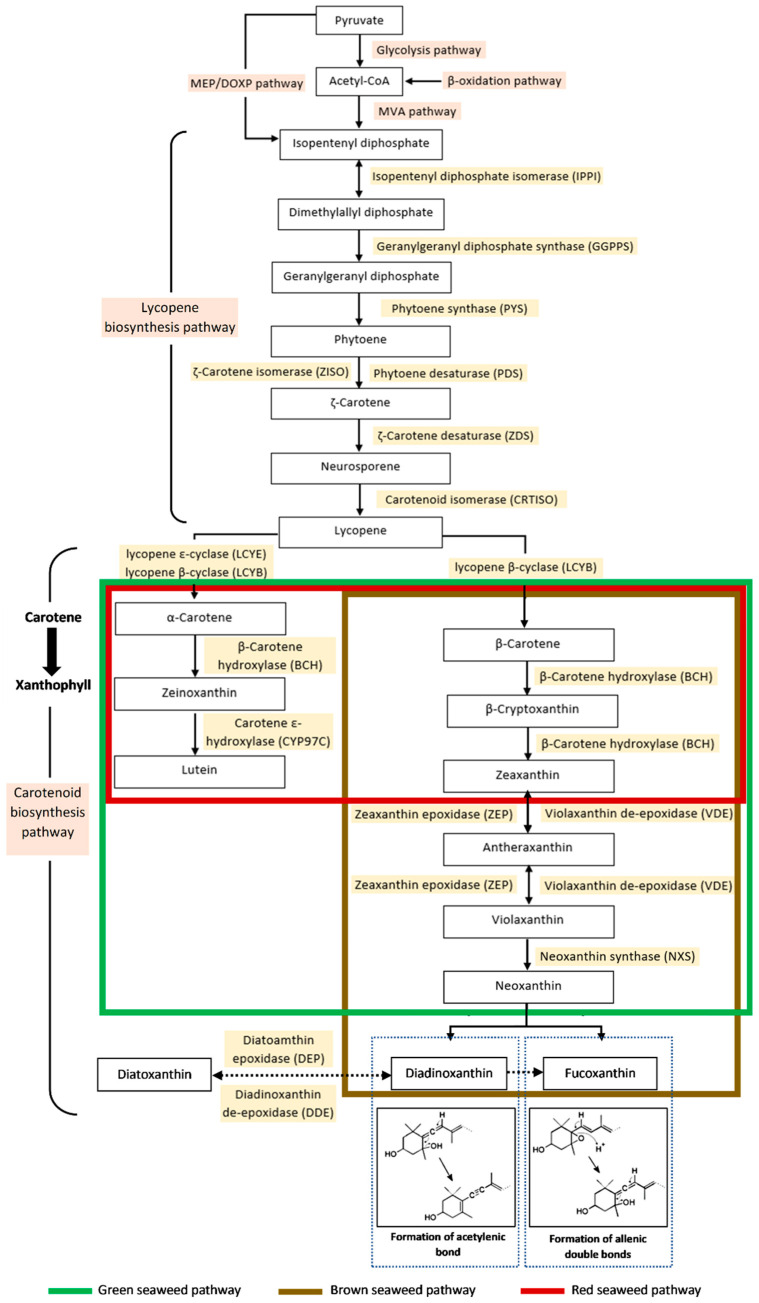
Carotenoid pathways in different seaweeds (Redrawn based on Mikami and Hosokawa [18] and Zarekarizi, Hoffmann and Burritt [36]).

**Figure 4 foods-11-02235-f004:**
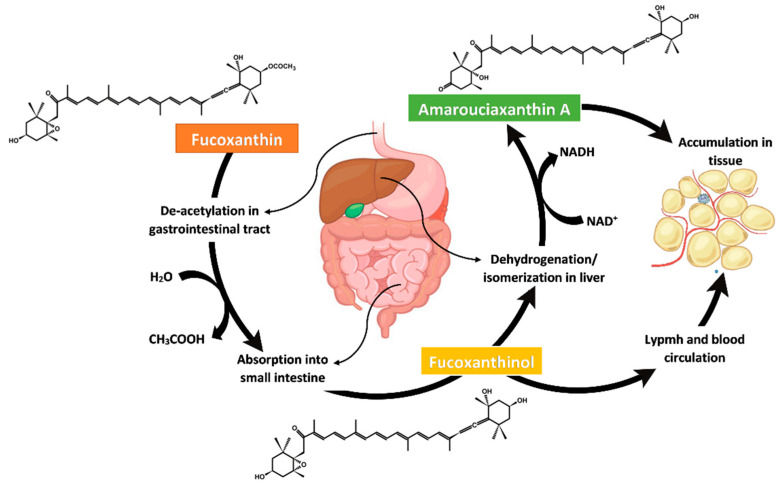
Metabolism of fucoxanthin.

**Table 1 foods-11-02235-t001:** Fucoxanthin content in various brown seaweed species.

Species	Fucoxanthin Content (mg/g Sample)	Sample Condition	Country Origin	Extraction Method; Solution	Detection Technique **	References
*Padina minor*	0.50	Dried	Malaysia	Solvent/maceration;ethanol	HPTLC	[37]
*Padina pavonica*	0.43					
*Sargassum polycystum*	0.41					
*Euchema cottoni*	0.94					
*Sargassum* sp.	1.50	Dried	Malaysia	Solvent/maceration;methanol	RP-HPLC-DAD	[38]
*Saccharina japonica*	0.07	Dried	Malaysia	Solvent/maceration;methanol	HPLC-DAD	[39]
*Sargassum siliquosum*	1.41	Dried	Malaysia	Solvent/maceration;methanol	HPLC-UV	[40]
*Sargassum polycystum*	0.31					
*Sargassum binderi*	7.4	Dried	Malaysia	Solvent/maceration;methanol	HPLC-UV	[41]
*Sargassum plagyophyllum*	0.71	Dried	Malaysia	Solvent/maceration;acetone-methanol	RP-HPLC-DAD	[42]
*Turbinaria turbinate*	0.59					
*Sargassum duplicatum*	0.146	Dried	Indonesia	Solvent/maceration;methanol	TLC	[43]
*Sargassum polycystum*	0.155 -0.587 *	Fresh	Indonesia	Ultrasound-assisted extraction (UAE); acetone	HPLC-UV	[44]
*Turbinaria decurrens*	0.65	Dried	Indonesia	Solvent/maceration;ethanol	HPLC-UV	[45]
*Sargassum crassifolium* J. Agard	1.64	Dried	Indonesia	Solvent/maceration;chloroform-methanol	HPLC-DAD	[46]
*Padina australis* Hauck	1.29					
*Turbinaria ornata* (Turner) J. Agardh	1.27					
*Sphaerotrichia divaricata*	0.11–1.48	Dried	Japan	Solvent/maceration;ethanol	HPLC-UV	[47]
*Sargassum horneri* (Turner) J. Agardh	2.12	Dried	Japan	Solvent/maceration;chloroform-methanol	HPLC-DAD	[46]
*Cystoseira Hakodatensis* (Yendo) Fensholt	1.99					
*Sargassum siliquastrum* (Mertens ex Turner) C. Agardh	2.01					
*Ecklonia kurome* Okamura	1.68					
*Undaria pinnatifida*	0.39	Fresh	Japan	Pressurized liquid extraction (PLE); liquefied dimethyl ether (DME)	HPLC-UV	[48]
*Sargassum horneri* (Turner)	1.35–4.49 *	Dried	Japan	Solvent/maceration;chloroform-methanol	HPLC-DAD	[49]
*Cystoseira hakodatensis* (Yendo)	0.63–4.14 *					
*Hincksia mitchellae* P.C. Silva	5.50	Dried	Taiwan	Soxhlet assisted extraction (SAE); ethyl acetate	HPLC	[50]
*Saccharina japonica* (Areschoug) C.E.Lane, C.Mayes, Druehl & G.W.Saunders	0.45	Dried	Korea	Soxhlet assisted extraction (SAE); n-hexane	HPLC	[51]
*Sargassum horneri* (Turner) C.Agardh	0.77	Dried	Korea	Supercritical fluid extraction (SFE); CO_2_-ethanol	HPLC-DAD	[52]
*Sargassum japonica* J.E. Areschoug	0.41					
*Sargassum horneri*	0.28	Fresh	China	Solvent/maceration;ethanol	HPLC-DAD	[53]
*Sargassum fusiforme* (Harvey) Setchell	2.62	Fresh	China	Solvent/maceration;acetone-ethanol	Spectro	[54]
*Laminaria japonica* Aresch	0.03	Fresh	China	Microwave-assisted extraction; Heptane-acetone-water	LC-ESI-MS, HPLC-UV, ^1^ H-NMR	[55]
*Sargassum fusiforme* (Harvey) Setchell	0.01	Dried				
*Undaria pinnatifida* (Harvey) Suringar	0.73					
*Cystoseira indica*	0.77–0.81	Dried	Iran	Ultrasound-assisted extraction (UAE); methanol (*C. indica*), methanol-acetone (*S. angustifolium*)	HPLC-DAD	[56]
*Sargassum angustifolium*	0.70–0.79					
*Dictyota indica*	0.211–0.463 *	Dried	Iran	Solvent/maceration;methanol	HPLC-UV	[35]
*Iyengaria Stellata*	0.026–0.055 *					
*Padina tenuis*	0.018–0.043 *					
*Colpomenia sinuosa*	0.014–0.019 *					
*Nizamuddinia zanardinii* (Schiffner) P.C.Silva	0.81–1.65 *	Dried	Iran	Solvent/maceration;chloroform-methanol-water	RP-HPLC-DAD	[33]
*Cystoseira indica* (Thivy & Doshi) Mairh	2.33–3.56 *					
*Sargassum swartzii C. Agardh*	0.17	Dried	India	Soxhlet assisted extraction (SAE); ethyl acetate	FTIR, ^1^ H-NMR, ^13^ C-NMR	[57]
*Padina tetrastromatica*	0.75	Dried	India	Ultrasonic-assisted extraction (UAE);ethanol	FTIR, HPLC-UV, Orbitrap-MS	[58]
*Dictyopteris australis*	0.23	Dried	India	Solvent/maceration;acetone	Spectro	[59]
*Dictyota dichotoma*	0.18					
*Iyengaria stellate*	0.18					
*Lobophora variegata*	0.23					
*Padina gymnospora*	0.43					
*Padina tetrastromatica*	0.41					
*Sargassum linearifolium*	0.37					
*Spatoglossum asperum*	0.58					
*Stoechospermum marginatum*	0.37					
*Turbinaria spp.*	0.43					
*Fucus vesiculosus*	0.657	Dried	Ireland	Solvent/maceration;water	HPLC-UV, LC-MS	[60]
*Alaria esculenta*	0.822					
*Himanthalia elongata*	18.6	Dried	Ireland	Solvent/maceration;hexane-diethyl ether-chloroform	LC-ESI-MS, HPLC-DAD, ^1^ H-NMR	[61]
*Alaria esculenta*	0.87	Fresh	Ireland	Solvent/maceration;acetone	HPLC-DAD	[62]
*Fucus vesiculosus*	0.7					
*Laminaria digitata*	0.65					
*Fucus serratus*	3.57	Dried	Ireland	Solvent/maceration;hexane-acetone	HPLC-DAD	[63]
*Laminaria digitata*	1.403					
*Ascophyllum nodosum* (Linnaeus) Le Jolis	0.022	Dried	Ireland	Vortex-assisted solid-liquid micro-extraction (VAE); ethanol	HPLC-PDA	[64]
*Fucus vesiculosus* Linnaeus	0.02					
*Dictyota dichotoma* (Hudson) J.V.Lamouroux	0.60	Dried	Portugal	Vortex-assisted solid-liquid micro-extraction (VAE); ethanol	HPLC-PDA	[64]
*Sargassum vulgare* C. Agardh	0.40					
*Zonaria tournefortii* (J.V.Lamouroux) Montagne	0.80					
*Laminaria ochroleuca*	0.004–0.16 *	Dried	Portugal	Solvent/maceration;acetone	HPLC-DAD	[32]
*Saccharina latissima*	0.02–0.12 *					
*Saccorhiza polyschides*	0.08–0.24 *					
*Sargassum muticum* (Yendo) Fensholt	0.55	Dried	Spain	Supercritical fluid extraction (SFE); CO_2_-ethanol	HPLC-DAD	[65]
*Cystoseira barbata* (Stackh.) C. Agardh	3.0	Dried	Ukraine	Solvent/maceration;ethanol	TLC	[66]
*Undaria pinnatifida*	0.70	Dried	New Zealand	Solvent/maceration;ethanol	HPLC-DAD	[67]
*Undaria pinnatifida*	1.77–2.08 *	Dried	New Zealand	Solvent/maceration;methanol	HPLC-DAD	[68]
	3.32–4.96 *	Fresh				

* Depend on seasonal changes. ** HPLC; High-performance liquid chromatography, RP-HPLC; Reversed-phase HPLC, HPTLC; High-performance thin-layer chromatographic, HPLC-DAD; HPLC with diode array detector, HPLC-PDA; HPLC with photo-diode array detector, HPLC-UV; HPLC with an ultraviolet detector, TLC; Thin layer chromatography, FTIR; Fourier-transform infrared spectroscopy, 1 H-NMR; Proton nuclear magnetic resonance, 13 C-NMR; Carbon-13 nuclear magnetic resonance, LC-ESI-MS; Liquid chromatography-electrospray ionisation-mass; Orbitrap-MS; Orbitrap mass spectrometer, LC-MS; Liquid chromatography-mass spectrometry, Spectro; Spectrophotometry.

**Table 2 foods-11-02235-t002:** Summary of fucoxanthin’s biological activities and its beneficial mechanisms.

Bioactivities	Mechanisms/Benefits	References
Antioxidant	High antioxidant activities detected by several antioxidant assays; DPPH, FRAP and ABTS Suppress the generation of reactive oxygen species (ROS), nitric oxide (NO), and lipid peroxidation in HaCaT cells, ARPE-19 cells, and RAW 264.7 macrophage cells and human skin melanoma (A375) cells Modulate Nrf2/ARE, ERK/p38, PI3 K/Akt, and Sirt1 signalling pathways Alter the ROS, glutathione (GSH), glutathione S-transferase (GST), catalases, HO-1, NQO1, and apoptosis-related protein production	[26,38,106,107,108,109,110,111,112,113,114,118,119,120,121,204,205]
Anti-inflammatory	Inhibit prostaglandin (PGE_2_) and NO production by downregulation of COX-2 and iNOS enzymes expression, respectively Prevent degradation of IκB-α phosphorylation and reduce ERK1/2, p38, and JNK MAPKs phosphorylation Attenuate TNF-α, MCP-1, IL-1β, and IL-6 production Photo-protective effects; ○ Downregulate inflammasome components ASC, caspase-1, NLRP3, and IL-1β ○ Prevent UVB-induced skin erythema and epidermal hyperplasia ○ Reduce the myeloperoxidase (MPO) activity, skin oedema, UVB-induced erythema and HO-1 protein upregulation Inhibit Th17 cell development and stimulate Foxp3^+^ Treg cell differentiation Improve the intestinal immune function and epithelial barrier against the lipopolysaccharide effect	[108,110,111,123,124,125,126]
Anti-obesity	Stimulate mitochondrial uncoupling protein 1 (UCP1) and promote β-oxidation Attenuate the leptin expression and increase adiponectin levels Inhibit pancreatic lipases Inhibit the glycerol-3-phosphate dehydrogenase action and downregulate peroxisome proliferator-activated receptor γ (PPARγ) Reduce HbA_1 c_ and glycated albumin levels Attenuate body and WAT weight and prevent excessive fat, lipid formation and adipocyte differentiation Decrease serum triglycerides level, plasma aminotransferase enzymes level, and blood pressure level Increase resting energy expenditure (REE) Downregulate mRNA levels of lipolysis-related genes (Lipe and Plin1), fatty acid uptake-related gene (Cd36), lipogenesis-related genes (Srebf1), and lipoprotein lipase coding (Lpl) Upregulate of the key transcriptional regulators (PPARγ, C/EBPα, and SREBP1 c) expression, transcription factors Regulate the adipogenic gene expression and a marker of adipocyte differentiation (aP2) Reduce PPAR, C/EBP, and SREBP1 c levels which inhibit IRS-1 phosphorylation Modulate SIRT-1 protein expression, AMPK, and FoxO pathways	[90,123,128,129,130,131,133,134,136,137,138,139,142,143]
Anti-diabetic	Reduce concentrations of plasma insulin, blood glucose, blood HbA_1 C_, and resistin levels Inhibit macrophage infiltration in both perigonadal and mesenteric WAT Decrease MCP-1 and TNF-α mRNA expression Attenuate overexpression of IL-6 mRNA and IL-6 generation Increase plasminogen activator inhibitor-1 (PAI-1) level and lessen expression levels of PAI-1 mRNA Decrease co-culture cells of 3 T3-L1 adipocyte and RAW264.7 macrophage cells Increase GLUT4 expression; elevation in translocation of GLUT4 to plasma membranes and improved EDL’s muscle translocation Increase the expression of IR mRNA by activating phosphorylation of Akt Upregulate PGC-1 α expression levels Stimulate serum adiponectin levels and decrease serum insulin levels Promote mRNA expression of the transcription factor peroxisome proliferator-activated receptor (PPAR) Inhibit the action of α -amylase hydrolyses oligosaccharides and α-glucosidase	[123,130,132,149,150,151,152,155,156,157,186]
Anti-cancer	Decrease numerous cancer cell viability Suppress the cell cycle in G0/G1, S, and/or G2/M phase depending on the cancer cell types Modulate several genes and protein expression, involving Mcl-1, STAT 3, p-STAT3, survivin, Bcl-2, Bcl-x, cIAP-2, XIAP, c-Myc, cyclin-dependent kinases (CDKs), and cyclin Induce apoptosis by altering several pathways; JAK/STAT signalling pathway, PI3 K/Akt/NF-κB signalling, and abruption of mortalin–p53 complex, and caspase activation Suppress fibroblast growth factor 2 (FGF-2) mRNA expression, receptor (FGFR-1), and trans-activation factor (EGR-1) Downregulates the phosphorylation of FGF-2-mediated intracellular signalling proteins (ERK1/2 and Akt) Reduce cells’ expression of PPAR and activation of Akt and increase the expression of integrin 1	[112,163,164,165,166,167,168,169,170,171,172,176]
Neuroprotective	Activate PI3 K/Akt cascade and inhibit ERK pathway Reduce the formation of Aβ plaques Suppress MAPK phosphorylation pathway Reverse the rise of acetylcholinesterase (AChE) activity Reduce choline acetyltransferase (ChAT) activity in the hippocampus and cortex Stimulate Nrf2- ARE and Nrf2-autophagy pathways and Nrf2/HO-1 signalling Exhibit mixed-type inhibition against BACE1; interact with BACE1 residues, Gly11 and Ala127	[113,149,180,181,182,183,184,185,186,188]
Antifibrotic	Attenuate the expression/production of α-smooth muscle actin (α-SMA), type 1 collagen (Col-1), fibronectin, and IL-6 Suppress MAPK phosphorylation, PI3 K/Akt pathway, Akt/SP-1 pathway, and Smad2/Smad3 pathway	[191,192]
Antitubercular	Suppress arylamine-N-acetyltransferase (TBNAT) and UDP-galactopyranose mutase (UGM)	[193]
Kidney protection	Upregulate Na+/H+ exchanger isoform 1 (NHE1) expression in renal tubules Inhibit renal fibrosis, reduced serum creatinine level, activated Akt, and inhibited H_2_ O_2_-induced apoptosis	[198,199,200,201]
Liver protection	Reduce liver weight gain, hepatic lipid oxidation, hepatic fat accumulation and mRNA expression levels of inflammation, and infiltration-related genes Suppress mRNA expression of lipogenesis-related genes, cholesterol esterification, lipid droplet accumulation, and induced CPT1 A mRNA level (β-oxidation related gene) Generate SREBP expression and reducing the liver’s cholesterol uptake through downregulation of SR-B1 and LDLR Accelerate omega-6 PUFA and omega-3 PUFA promotion to arachidonic acid (AA) and docosahexaenoic acid (DHA)	[111,202,203,206]

**Table 3 foods-11-02235-t003:** Summary of different food products incorporated with fucoxanthin as functional ingredients.

Food	Fucoxanthin Source	Type of Source	Sensory Acceptability	Fucoxanthin Total Lost	References
Shrimp paste	*Sargassum* sp. extract	Macroalgae	Favourable at 12%	N/A	[212]
Fortified skimmed and whole milk	Food grade fucoxanthin	N/A	N/A	4%	[64]
Dried tea	*Sargassum cristaefolium* extract	Macroalgae	N/A	N/A	[213]
Plain yogurt	*Pavlova lutheri* lipid extract	Microalgae	Not acceptable	N/A	[214]
Fortified skimmed and whole milk	*Phaeodactylum tricornutum* extract	Macroalgae	N/A	9%	[215]
Yogurt	*Fucus vesiculosus* and *Ascophyllum nodosum* extract	Macroalgae	Favourable at 0.5%; water extract of *A. nodosum*	0%	[216]
Fluid milk	*Fucus vesiculosus* and *Ascophyllum nodosum* extract	Macroalgae	Not acceptable	0%	[216]
Scones	*Undaria pinnatifida* powder	Macroalgae	Favourable at 0.5 and 2%	<15%	[217]
Semolina wheat-based pasta	*Undaria pinnatifida* powder	Macroalgae	Favourable at 10%	<10%	[218]
Ground chicken breast meat	*Undaria pinnatifida* extract	Macroalgae	N/A	N/A	[219]

## Data Availability

Not applicable.

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
