# Peer review of "Brown Algae as Functional Food Source of Fucoxanthin: A Review"

_foods, 2022, doi:10.3390/foods11152235_

Round 1
Reviewer 1 Report
This review describes the role of algal fucoxanthin as functional food supplement. The health benefits of fucoxanthin along with the mode of actions are discussed clearly. The topic addressed in this review is scientifically interesting one due to the growing needs for functional foods. Though the authors prepared a detailed review, I think this manuscript needs to be shortened with more focus on the title instead of going for overall benefits of fucoxanthin. My suggestions for revising the manuscript.
1. Provide more tables for explaining potential biological activities of fucoxanthin. This will reduce the text considerably.
2. Try to include a figure or flow diagram for explaining the mode of action of antidiabetic activity of fucoxanthin
Reviewer 2 Report
The article is important and interesting as an overview of the medical, biotechnological and food direction of the practical use of seaweeds and their importance for humans. However, the materials of the article require minor corrections, individual additions and updates.
L. 40: Fucoxanthin is a carotenoid characteristic of seaweed, in particular brown algae. However, among this taxonomical group of algae there are also freshwater organisms that grow outside the marine environment, but have this carotenoid (characteristic of representatives of Phaeophyta), and therefore the phrase (L. 40) "...xanthophyll of marine carotenoid" must be replaced.
L. 40, 166-167, 171: The authors need to distinguish between the concepts of "taxonomic units", "taxonomic subdivisions" and their taxonomic rank and group size and non-taxonomic groups of organisms (line. 40). The use of the endings “-phyta” attests to the division rank (which the authors operate with), and the class rank - classis has the ending of the taxonomic group “-phyceae” (compare L. 40, 166-167, 171, etc.).
L. 166-167 … Each seaweed "classification"… - it is not "classification", but "taxonomical group".
The use of microalgae and their cultivation in bioreactors is an industrially and economically profitable process, and therefore the unequivocal conclusion / statement (L. 123) "commercial production of the former is not practical as it requires" needs correction.
L. 148 Table 1. Fucoxanthin content in various brown seaweed species:
it is necessary to indicate the authors of the species-producers of fucoxanthin (according to cited literature sources or modern databases of algae) in order to understand the taxonomic scope of the species and its current presentation - perception as an object of biotechnological research.
The article is important and interesting as an overview of the medical, biotechnological and food direction of the practical use of seaweeds and their importance for humans. However, the materials of the article require minor corrections, individual additions and updates.
L. 40: Fucoxanthin is a carotenoid characteristic of seaweed, in particular brown algae. However, among this taxonomical group of algae there are also freshwater organisms that grow outside the marine environment, but have this carotenoid (characteristic of representatives of Phaeophyta), and therefore the phrase (L. 40) "...xanthophyll of marine carotenoid" must be replaced.
L. 40, 166-167, 171: The authors need to distinguish between the concepts of "taxonomic units", "taxonomic subdivisions" and their taxonomic rank and group size and non-taxonomic groups of organisms (line. 40). The use of the endings “-phyta” attests to the division rank (which the authors operate with), and the class rank - classis has the ending of the taxonomic group “-phyceae” (compare L. 40, 166-167, 171, etc.).
L. 166-167 … Each seaweed "classification"… - it is not "classification", but "taxonomical group".
The use of microalgae and their cultivation in bioreactors is an industrially and economically profitable process, and therefore the unequivocal conclusion / statement (L. 123) "commercial production of the former is not practical as it requires" needs correction.
L. 148 Table 1. Fucoxanthin content in various brown seaweed species:
it is necessary to indicate the authors of the species-producers of fucoxanthin (according to cited literature sources or modern databases of algae) in order to understand the taxonomic scope of the species and its current presentation - perception as an object of biotechnological research.
Reviewer 3 Report
This manuscript provides an overview on fucoxanthin as bioactive compound from a food source, brown algae. Biosynthesis pathway of fucoxanthin and its derivatives is also provided. The biological activities of fucoxanthin including antioxidant, anti-inflammatory, anti-obesity, antidiabetic, anticancer activities, as well as other activities. The mechanism of action of individual activity is also provided. Information of fucoxanthin as functional food ingredient is given, and the stability of fucoxanthin in food products is provided. In general, the content of this review is useful for readers. In order to improve this manuscript, please consider the comments and suggestions, which are listed below.
1. Please consider the title “Brown Algae as Functional Food Source of Fucoxanthin: A Review”. The aim is for functional food ingredient, so this title may be appropriate.
2. Keywords should contain the words “Marine natural product”; “Functional foods”
3. “Brown seaweed is typically a complex polysaccharide with high alcohol content.”; would it be “Brown seaweed has a complex polysaccharide with high alcohol content.”? Please consider.
Reviewer 4 Report
In this study, the authors aim to review the studies regarding fucoxanthin from brown seaweed, its biological activities, and functional food potential.
Line 42: “Brown seaweed is typically a complex polysaccharide with high alcohol content.”
This sentence should be revised because it is wrong. Brown algae is not a polysaccharide. It should be revised in a way that brown algae contain complex polysaccharides with high alcohol content.
Line 72: “The bioactive compound of carotenoids is present in all algae as they represent photosynthetic pigments implicated for the red, yellow, and orange hues”
The sentence has no meaning, and it is complex. This should also be revised since there is a wrong meaning. As a suggestion:
Carotenoids as bioactive compounds are present in all algae as they represent photosynthetic pigments responsible for the red, yellow, and orange colors.
Line 88: Figure 1 has some mistakes. It should not be written as pure hydrocarbons. In addition, Pure hydrocarbon’s structure contains epoxy oxygen which is wrong. It should be corrected. Carotenes do not contain oxygen. My other suggestion is that the authors should better add some more carotenes as structures like lycopene.
Line 175: “Figure 3. Fucoxanthin biosynthesis pathway in different seaweed”
The caption for Figure 3 should be revised. Only brown algae produce fucoxanthin. Green and red seaweed produce other carotenoids. It can be alternatively, Carotenoid pathways in different seaweeds.
Line 855: “Algae is one of the established sources of fucoxanthin, a photosynthetic component, and a very valuable marine carotenoid”
This sentence is wrong unless it should be written "brown algae" at the beginning of the sentence. Fucoxanthin is a carotenoid synthesized in brown algae and it should be emphasized, that it cannot be used as other carotenoids by generalizing that all algae can produce it.
Line 874: “It is an approach to make fucoxanthin a more sustainable alternative to many other carotenoids bioactive compounds such as lutein and β-carotene.”
I disagree with the idea in this sentence. Fucoxanthin cannot be an alternative to other carotenoids. It can be told that fucoxanthin is also a promising carotenoid like lutein and β-carotene and it should get the attention it deserves in the food market or pharmaceutical industry.
Round 2
Reviewer 1 Report
The authors improved the manuscript. It seems to be better than the last version.